# State-of-the-Art Review on Botanical Hybrid Preparations in Phytomedicine and Phytotherapy Research: Background and Perspectives

**DOI:** 10.3390/ph17040483

**Published:** 2024-04-10

**Authors:** Alexander Panossian, Terry Lemerond, Thomas Efferth

**Affiliations:** 1Phytomed AB, Sjöstadsvägen 6A, Lgh 1004, 59344 Västervik, Sweden; 2EuroPharma USA Inc., Green Bay, WI 54311, USA; terryl@europharmausa.com; 3Department of Pharmaceutical Biology, Institute of Pharmaceutical and Biomedical Sciences, Johannes Gutenberg University, 55099 Mainz, Germany

**Keywords:** network pharmacology, gene expression, botanical hybrid preparations, *Rhodiola rosea* clinical trials, synergy

## Abstract

Background: Despite some evidence supporting the synergy concept, the commonly known assumption that combinations of several herbs in one formulation can have better efficacy due to additive or synergistic effects has yet to be unambiguously and explicitly studied. Study aim: The study aimed to reveal the molecular interactions in situ of host cells in response to botanical hybrid preparations (BHP) intervention and justify the benefits of implementing BHP in clinical practice. Results: This prospective literature review provides the results of recent clinical and network pharmacology studies of BHP of *Rhodiola rosea* L. (Arctic root) with other plants, including *Withania somnifera* (L.) Dunal (ashwagandha), (*Camellia sinensis* (L.) Kuntze (green tea), *Eleutherococcus senticosus* (Rupr. and Maxim.) Maxim. (eleuthero), *Schisandra chinensis* (Turcz.) Baill. (schisandra), *Leuzea carthamoides* (Willd.) DC., caffeine, *Cordyceps militaris* L., *Ginkgo biloba* L.(ginkgo), *Actaea racemosa* L. (black cohosh), *Crocus sativus* L. (saffron), and L-carnosine. Conclusions: The most important finding from network pharmacology studies of BHP was the evidence supporting the synergistic interaction of BHP ingredients, revealing unexpected new pharmacological activities unique and specific to the new BHP. Some studies show the superior efficacy of BHP compared to mono-drugs. At the same time, some a priori-designed combinations can fail, presumably due to antagonistic interactions and crosstalk between molecular targets within the molecular networks involved in the cellular and overall response of organisms to the intervention. Network pharmacology studies help predict the results of studies aimed at discovering new indications and unpredicted adverse events.

## 1. Introduction

The use of complex herbal formulations comprising fixed combinations of several plant extracts has a long history in TCM, Kampo, Ayurveda, and other traditional medical systems [1,2,3]. The potential health benefits of consuming combined nutrients or dietary supplements have gained considerable attention due to their impact on overall well-being due to the synergy concept, recently defined as nutrient synergy [4]. The term synergy is differently defined in the various scientific disciplines and comes from the Attic Greek word συνεργία synergia [1] from *synergos*, συνεργός, meaning “working together”.

Combining two or more plants assumes that a hybrid botanical preparation (BHP) is more active due to synergistic effects brought about by different mechanisms. Figure 1 illustrates an allegoric analogy with two or more kinds of hybrids from ancient mythology.

*Lama* an Assyrian protective hybrid deity with a human head, a bull’s body or lion’s body, and wings symbolizing the synergistic interaction between these elements.The centaur Chiron, from Greek mythology, raised Achilles at the request of Achilles’ mother (reproduced from Peter-Paul Rubens in 1630–1635).*Navagunjara*—A Hindu creature with the head of a rooster, the neck of a peacock, the back of a bull, a snake-headed tail, three legs of an elephant, tiger, and deer or horse, with the fourth limb being a human hand holding a lotus.*Nureonna*—a creature with the head of a woman and the body of a snake (Japanese mythology).*Kotobuki*—a Japanese Chimera with the head of a rat, the ears of a rabbit, the horns of an ox, the comb of a rooster, the beard of a sheep, the neck of a Japanese dragon, the mane of a horse, the back of a wild boar, the shoulders and belly of a South China tiger, the arms of a monkey, the hindquarters of a dog, and the tail of a snake.

Based on the assumption of synergistic interaction of several components, researchers propose that combinations of several active ingredients in one formulation can have superior effectiveness and better efficacy due to their multiple effects on various targets (Figure 2) [5,6,7,8,9,10].

The term botanical “Hybrid” preparation (BHP) is coined to describe the biological/pharmacological activity (conditional pharmacological “signature”) of a fixed herbal combination with a specific chemical composition (e.g., TLC of HPLC conditional chemical “fingerprint”) (Figure 2) [7]. BHP is like a “newborn” unique biologically active substance, a hybrid of the “parent” ingredients [7].

Similarly, phytochemical hybrid preparation (PHP) is used to determine the pharmacological signature of a combination of phytochemicals comprising specific plant species, e.g., *R. crenulata* (Hook. f. and Thomson) H. Ohba, *R. sacra* (Prain ex Raym.-Hamet) S.H. Fu, *R. kirilowii* (Regel) Maxim., *R. quadrifida* (Pall.) Fisch. & C. A. Mey., and *R. dumulosa* (Franch.) S. H. Fu, etc., characterized by specific chemical fingerprints (Figure 2).

This distinction emphasizes that any new fixed combination exhibits unique biological characteristics and effects different from the ingredients’ natural characteristics. That is due to their multitarget effects on various mediators, which interact within various regulatory systems of the host cells and organism [7,8].

Modern technologies in biomedical research and bioinformatics provide potent tools in phytotherapy research and implement a concept of systems biology and network pharmacology, uncovering numerous molecular targets and new mechanisms of action of botanical (Figure 3) [11]. In these studies, the experimental protocol included mRNA microarray hybridization, ingenuity pathway analysis (IPA), and statistical analysis to uncover the mechanism of action of purified compounds, the plant extract, and their combinations comprising BHP. This was achieved by assessing their effects on gene expression in isolated neuronal cells and their potential therapeutic action.

Recently, the Herbal Medicinal Products Platform Austria (HMPPA) committee chose *Rhodiola rosea* L. as a medicinal plant in 2023 in Austria [12]. What is behind this choice of HMPPA?

*Rhodiola rosea* L. (Crassulaceae, syn. *Sedum rhodiola* DC., *Sedum rosea* (L.), Scop cop, known as Roseroot, Rosenroot, Golden Root, Arctic Root, Orpin Rose, Rhodiole, and Rougeâtre) has an extensive history as a treasured medicinal plant and has appeared in the Materia Medica of several European countries [13,14]. In Europe, Rosenroot was formally adopted in Sweden as a natural remedy (national legislation) from 1987 to 2008, and since 2008, as a traditional herbal medicinal product (THMP) and registered as an adaptogen in decreased performance, such as fatigue and weakness [15].

During the last two decades, more than 1200 studies, including 33 clinical and 910 pre-clinical studies, were conducted in Europe, America, and China, providing results of preclinical and clinical efficacy, safety, and quality of *R. rosea* preparations in various stress-induced disorders, including fatigue syndrome, cognitive deficiencies, mild/moderate depression, anxiety, and burnout symptoms, as well as in healthy subjects under stress (Figure 4) [15,16,17,18,19,20,21,22,23,24,25,26,27,28,29,30,31,32,33,34,35,36,37,38,39,40,41,42,43,44,45,46,47,48].

Eighteen clinical studies were conducted on the fixed combinations of *R. rosea* with green tea [49,50,51,52,53], Eleutherococcus, and Schisandra [54,55,56,57], Eleutherococcus, Schisandra, and Leuzea [56], caffeine [58,59], Cordyceps [60,61,62,63], Ginkgo [64], black cohosh [65], Saffron [66], L-carnosine [67], Eleutherococcus, and Glycyrrhiza [68].

This narrative state-of-the-art review provides the results of recent clinical [49,50,51,52,53,54,55,56,57,58,59,60,61,62,63,64,65,66,67,68] and network pharmacology studies of the BHP of *R. rosea* with other plants [5,6,8,9,10]. The studies aim to reveal the molecular interactions in situ of host cells in response to the intervention of BHP and justify the benefits of implementing BHP in clinical practice.

## 2. Synergy and Antagonism of Active Ingredients of *Rhodiola rosea* and Other Plant Extracts

Concomitant treatment of disease by two herbal drugs aiming to achieve better effect suggests their additive (1 + 1 = 2), amplifying (1 + 1 > 2), potentiation (0 + 1 > 1), or synergistic (0 + 0 > 1) result. Hypothetically, their interaction can result in beneficial antagonistic interactions (1 + 1 < 2, or 1 + 1 = 0), e.g., decreased toxicity and detoxifying effects [3,6].

This hypothesis was verified in a set of in vitro studies, where the effects of several BHP and their ingredients on the number of deregulated genes in brain cell cultures were analyzed [5,6,7,8,9,10]. The composition of genes (signature) deregulated by BHP was quantitatively and qualitatively different from the signature (composition of genes) deregulated by each plant separately, suggesting that the impact of the BHP on the target cells was qualitatively different from the effects of individual ingredients [5] (Figure 5). This implies that the BHP exhibits quite different pharmacological activities when the ingredients are combined. These findings are essential for understanding the unpredictable results of clinical studies of multi-component drugs and dietary supplements [49,50,51,52,53,54,55,56,57,58,59,60,61,62,63,64,65,66,67,68].

Figure 5 and Figure 6 illustrate the essence of hybridization of ingredients of BHP or PHP, their synergy, and the antagonism in these experiments and interpretations. The point is the biological activity of a single compound, e.g., salidroside, an active compound of *Rhodiola rosea* extract, interacts with many proteins in brain cells, deregulating 640 (!) genes in neuroglia cells, associated with various physiological processes and effects in stress and aging-induced disorders (e.g., neurodegeneration). This is illustrated in Figure 5 and Figure 6 [9].

Meanwhile, the total extract of *Rhodiola*-containing 120 phytochemicals (including salidroside) or the BHP of *Rhodiola, Schisandra*, and *Eleutherococcus* extracts (ADAPT-232) containing 207 phytochemicals deregulate almost the same number of genes (*Schizandra*—625 genes, *Rhodiola*—631 genes, *Eleutherococcus*—669 genes, ADAPT232—678 genes) [6] (Figure 7). Among those deregulated by ADAPT-232, there were 206 genes that were not deregulated by any ingredient of BHP ADAPT-232 due to the synergy effect (Figure 7). The synergy-derived biological effect is characteristic of the BHP (ADAPT-232), which has a distinct pharmacological profile (signature) and typical chemical composition (fingerprint), which are different from *Rhodiola rosea* extracts [5].

Gene expression profiling was conducted on the human neuroglial cell line, T98G, after treatment with either Rhodiola SHR-5 extract or several of its constituents separately, including salidroside, triandrin, and tyrosol. Rhodiola SHR-5 and individual constituents had similar effects on G-protein-coupled receptor (GPCR)-mediated signal transduction through cAMP, phospholipase C, and phosphatidylinositol signaling pathways (Figure 6).

The interpretations of microarray data and gene expression changes were conducted using Ingenuity Pathways Analysis (IPA) software (QIAGEN Bioinformatics, Aarhus C, Denmark), which is based on a continuously updated database (the Ingenuity Knowledge Base), gathering research observations with more than 8.1 million findings manually curated from biomedical literature and integrated from 45 third-party databases. The IPA network contains more than 40,000 nodes representing human genes, molecules, and biological functions, linked by 1,480,000 edges representing experimentally observed cause–effect relationships (either inhibiting or activating) associated with gene expression, transcription, molecular metabolism, and receptor binding. Network edges are also linked to activating or inhibiting effects. The IPA core analysis of transcriptomic datasets provides information about the impact of test samples on canonical signaling and metabolic pathways, diseases, and molecular and cellular functions that are activated or inhibited in experiments.

Two statistical methods of analysis of gene expression datasets are used in the IPA: (i) the gene-set-enrichment method, where differentially expressed genes are intersected with sets of genes that are associated with a particular pathway or biological function, providing a so-called “enrichment” score (Fisher’s exact test *p*-value). This score measures the overlap of the observed and predicted regulated gene sets. (ii) The method based on cause–effect relationships related to the direction of effects reported in the literature, which provides the so-called z-score measuring the match of observed and predicted up/down-regulation [10,11]. The predicted effects are based on gene expression changes in the experimental samples relative to the control; z-score > 2, −log *p*-value > 1.3.

Figure 7 shows the synergy, potentiation, and antagonistic effects of hybridization of a combination of *Rhodiola* with *Withania*, *Withania* with melatonin, and *Curcuma longa* with *Boswellia* on eicosanoids signaling pathways, which play an important role in inflammation and neurodegeneration in neuroglia cells.

In a recent study [8], BHP of *Rhodiola* with *Withania* (Adaptra) positively regulated 22 of 57 genes, which are known to activate the development of neurons (Figure 8), suggesting that Adaptra is potentially helpful in learning and memory, stress, and depression, insomnia, and aging-related neurodegenerative diseases, preventing Alzheimer’s and Parkinson’s diseases, and aiding recovery from brain injury and stroke.

Notably, 25 genes were deregulated due to RR and WS synergistic interactions in the fixed combination Adaptra (Figure 8 and Table 1). This means that in combination, these two ingredients of Adaptra act synergistically. In other words, Adaptra is superior to *Rhodiola* or *Withania* in the activation of neurogenesis and, consequently, has the potential effects mentioned above [8]. 

Figure 9 shows the Venn diagrams of deregulated genes induced by the treatment of neuroglial cells with *Withania somnifera* (WS), *Rhaponticum cartamoides* L. (RC), and *Eleutherococcus senticosus* (RS) root extracts, as well as their hybrid combination (RC-ES-WS) [6].

## 3. Clinical Studies in Human Subjects

### 3.1. BHP of Rhodiola with Green Tea (Mg-Teadiola^®^) in Psychological and Social Stress

Green tea contains catechins, tannins, phenolic acids, flavanol glycosides, the alkaloid caffeine, and the amino acid L-theanine [69,70], which are known to be capable of significant effects on the general state of mental alertness or arousal [71], activating adaptive cellular stress responses, inducing the production of cytoprotective proteins, and protecting neurons in animal models of Parkinson’s disease, Huntington’s disease, Alzheimer disease, and ischemia–reperfusion injury [72,73]. Green tea components, such as epigallocatechin gallate (EGCG), flavonoids kaempferol, and genistein activate protective mechanisms, including antioxidant and detoxifying enzymes via activation of Nrf2 signaling pathway, and upstream PKC, PI3K, and MAPKs modulation [70,72,73].

Both *Rhodiola rosea* and green tea (*Camelia sinensis*) supplementation were known to improve subjective stress perception and mood responses to acute stress. Their combinations with magnesium, vitamins B6, B9, B12, and L-theanine in a BHS Mg-Teadiola^®^ was developed by Sanofi-Aventis Group, France, and studied in two clinical trials conducted in France and the UK to assess stress-protective effects compared to the efficacy of *Rhodiola rosea* and green tea [49,50,51,52,53], as shown in Table 2. The authors hypothesized that the efficacy of BHS Mg-Teadiola^®^ is superior to that of the ingredients and/or placebo. In a DB-R-PC-PG clinical trial (NCT03262376), the single dose effect of Mg-Teadiola^®^ tablets, *Rhodiola*, and green tea extracts was studied in four parallel groups of 100 moderately stressed, otherwise healthy volunteers (DASS score: 13–25) after acute psychological and social stress experimentally induced by The Trier Social Stress Test (TSST, speech, and mental mathematics tasks). The outcome measures were as follows: (i) spectral theta brain activity associated with cognitive task performance, (ii) subjective stress (stress and arousal), (iii) mood (profile of mood states), (iv) salivary cortisol, and (v) cardiovascular parameters (BP, HRV) [49,50,51,52,53]. 

BHS Mg-Teadiola^®^ significantly alleviated subjective stress and mood responses to acute stress; analyses supported the superiority of the BHS Mg-Teadiola vs. placebo and the ingredients—*Rhodiola* and green tea. The BHS Mg-Teadiola^®^ significantly attenuated subjective stress, tension, and total mood disturbance ratings after acute stress exposure. These effects were found both during the peak stress response and recovery. The salivary cortisol response was unaffected by treatment [49]. The BHS Mg-Teadiola^®^ treatment significantly increased EEG resting state theta activity—considered indicative of a relaxed, alert state, attenuated subjective stress, anxiety, and mood disturbance, and heightened emotional and autonomic arousal; Mg-Teadiola may enhance coping capacity and offer protection from the harmful effects of stress exposure [50].

The BHS Mg-Teadiola^®^ increased spectral theta brain activity during the execution of two attentional tasks, suggesting a potential to increase attentional capacity under stress conditions [51].

In a placebo-controlled randomized clinical trial, the repeated dose effect of Mg-Teadiola^®^ tablets for four weeks was studied in two parallel groups of 100 moderately chronically stressed, otherwise healthy volunteers (the Depression Anxiety Stress Scale-42 score: >14) and two groups of 40 individuals after acute thermal stimulation. The outcome measures were as follows: (i) blood-oxygen-level-dependent signal, (ii) stress, (iii) depression, (iv) anxiety, (v) salivary cortisol, and (vi) sleep [52].

Mg-Teadiola^®^ was effective in relieving stress on days 14 and 28 in chronic stress and may diminish pain perception, underlining its potential benefits for patients suffering from pain, in whom comorbidities such as stress and sleep disorders are frequent [42]. Supplementation with Mg-Teadiola^®^ reduced stress on D28 in chronically stressed but otherwise healthy individuals and modulated the stress and pain cerebral matrices during stressful thermal stimulus [53].

### 3.2. BHP of Rhodiola SHR-5 with Schisandra and Eleutherococcus (ADAPT-232/Chisan^®^) for Relief of Mental and Physical Fatigue Both in Healthy Subjects and in Patients with Pneumonia and COVID-19

ADAP232/Chisan is the combination of extracts of *Rhodiola rosea* L., roots (SHR-5) *Eleutherococcus senticosus* (Rupr. et Maxim) Harms, roots (SHE-3) and *Schisandra chinensis* (Turcz) Baill., fruits, containing 0.5% schizandrin, 0.47% salidroside, 0.59% rosavin, 0.11% eleutherosides B and 0.19% E. ADAPT 232^®^ capsules and Chisan^®^ oral solution have been used to enhance mental and physical capacities in case of tiredness or during convalescence, as a natural remedy in Sweden since 1979 and as an herbal medicinal product in Denmark since 2002. In 2008, it was approved as a traditional herbal medicinal product in 2008 in Sweden as an adaptogen in case of decreased performance, such as fatigue and sensation of weakness.

Early studies of ADAPT-232 showed improved cognitive functions and endurance under the stress of the cosmonauts. It has been used in space and by cosmonauts during the longest flight in space in the 1990s. It is noteworthy that typically, only carefully verified aids, validated for their efficacy and safety, were allowed for use in space. ADAPT-232 showed a good anti-fatigue effect, significantly increasing accuracy and precision in psychometric tests, particularly in complicated tests, decreasing the number of errors and improving short memory in fatigue under space-like conditions. Significant improvements in concentration, oculomotor coordination, and short-term memory of cosmonauts have been demonstrated in the computerized Monotonic 3 test for attention, as well as in two complicated short-term memory tests [74].

Three randomized, placebo-controlled, double-blind clinical trials (RCT) were conducted on healthy subjects, and two RCT studies were conducted to assess the anti-fatigue effect of ADAPT232/Chisan during recovery of patients with infectious respiratory diseases [54,55,56,57]. The efficacy of a single dose of ADAPT-232 on cognitive functions in humans has been demonstrated in the participants who experienced stressful cognitive tasks, namely, the Stroop Color-Word test followed by the d2 test of attention (d2), before and two hours after treatment. The results of the d2 test (attention deficit) are shown on the graph below [55], Figure 10a. ADAP232/Chisan increases tolerance to mental fatigue in healthy subjects [55]. ADAP232 significantly improved attention and increased speed and accuracy during stressful cognitive tasks compared to placebo [55]. ADAP232 significantly decreased the number of mistakes in complicated psychometric tests [54,55,56,57] and improved patients’ quality of life and recovery from acute nonspecific pneumonia [54], as shown in Figure 10b. ADAPT232/Chisan has beneficial effects on stress-induced fatigue [55] in patients during their recovery from pneumonia [54] (Figure 10b).

ADAP232 and the BHP, containing *Rhodiola, Eleutherococcus, Schisandra*, and *Leuzea* (*Rhaponticum cartamoides* L.) roots extracts, significantly decreased inattention, impulsivity, and the perception of stress, while also reducing fatigue and increasing the anabolic index (testosterone/cortisol ratio) (Figure 10c–f) in 200 elite athletes. Furthermore, these supplements improved physical performance and the recovery rate of athletes after heavy physical and emotional load, indicating increased adaptation to physical and emotional stress [56].

Recently, the effects of ADAPT-232 on the recovery of patients with long COVID symptoms (fatigue, headache, respiratory insufficiency, cognitive performance, mood disorders, loss of smell, taste, and hair, sweatiness, cough, and pain in joints, muscles, and chest) were studied [59]. One hundred patients confirmed positive with the SARS-CoV-2 test, who were discharged from the intensive care unit and experienced at least three of nine long COVID symptoms in the 30 days before randomization, were included in the study.

ADAPT-232 reduced the duration of pain and fatigue by two days and one day, respectively, in 50% of patients (Figure 10h,i). The rate of cases of fatigue and pain symptoms was significantly lower in the rehabilitation period in the ADAPT-232 treatment group than in the placebo group on days 9 (39% vs. 57%, pain relief, *p* = 0.0019) and 11 (28% vs. 43%, relief of fatigue, * *p* = 0.0157). ADAPT-232 showed a significant increase in physical activity, measured as daily walk time (a workout), compared to placebo (Figure 10g). Clinical testing of C-reactive protein and the blood coagulation marker D-dimer did not show any statistically significant difference over time in the treatment between groups, with the exception of significantly reduced inflammatory cytokine IL-6 in the ADAPT-232 treatment group. In addition, ADAPT-232 significantly decreased blood creatinine compared to the placebo, suggesting prevention of the progression of possible renal failure in patients with long COVID [57].

### 3.3. BHP of Rhodiola with Caffeine for Enhancing Muscle Strength and Muscular Endurance

Alkaloid caffeine is a central nervous system (CNS) stimulant widely consumed in coffee and tea, promoting athletic capacities, including anaerobic exercise capacity, strength, and muscular and aerobic endurance. From 1984 to 2004, the World Anti-Doping Agency (WADA) banned caffeine because it enhances athletic performance in sports such as running, swimming, and cycling, but lifted the ban in 2004, removing caffeine from the list of prohibited substances, despite growing evidence that it is a sports booster. Athletes usually receive 3–6 mg/kg doses of caffeine to enhance resistance exercise performance acutely. However, a dose above 9 mg/kg often fails to cause a positive impact on various performance improvements and may also lead to side effects, such as tachycardia, headache, and anxiety. Long-term caffeine intake leads to an upregulation of adenosine receptor expression, resulting in a progressive reduction of the stimulatory effect on adenosine receptors and the development of caffeine resistance in the human body, which gradually occurs after 28 days of long-term caffeine intake [58]. Furthermore, regular use of caffeine-containing beverages may develop a physical, emotional, and psychological dependence, and one may experience a caffeine withdrawal syndrome characterized by headache, anxiety, irritability, low energy levels, dizziness or light-headedness, mental fogginess, and negative mood after abrupt cessation of caffeine intake. Unlike other psychoactive drugs, caffeine is legal, cheap, and unregulated in almost all parts of the world [75,76].

In two recent studies, the stimulating effects of caffeine, *Rhodiola rosea*, and their concomitant supplementation were compared in healthy human subjects and rats, suggesting the superimposed effects of the two supplements due to their different application mechanisms [58,59]. The impact of caffeine (400 mg/daily), *Rhodiola rosea*, (2.4 g/daily), and their combination for 30 days on muscle strength and muscular endurance physical performance of resistance exercise-untrained volunteers, as well as resistance exercise-trained subjects was assessed in four groups (12 participants in each group) of resistance exercise-trained subjects and two groups of resistance exercise-untrained volunteers, as shown in Table 2 [58]. *Rhodiola*-caffeine supplementation improved the rat model’s forelimb grip strength, erythropoietin, dopamine, and oxygen consumption rate. *Rhodiola*-caffeine significantly increased the bench press one-repetition maximum, deep squat 1RM, maximum voluntary isometric contraction, and maximum repetitions at 60% 1RM bench press in resistance exercise-untrained volunteers. BHP *Rhodiola*-caffeine improved resistance exercise performance significantly by increasing the bench press one-repetition maximum, deep squat 1RM, maximum voluntary isometric contraction, and maximum repetitions at 60% 1RM bench press for both resistance exercise-untrained and -trained volunteers. The authors concluded that the stimulating effect of *Rhodiola*-caffeine was superior to *Rhodiola* and caffeine separately, presumably due to their synergistic and different mechanisms of action [58].

In the follow-up open-label placebo-controlled study [59], depicted in Table 2, the participants without prior aerobic training experience in the *Rhodiola*-caffeine group showed significant improvements compared to the placebo group in maximal oxygen consumption (VO2max), 5 km run, countermovement jump (CMJ), standing long jump, and 30 m sprint. For individuals with years of aerobic training experience, the *Rhodiola*-caffeine group exhibited similar results, enhancing performance in the training exercise, except for a 30 m sprint, compared to the placebo group. The authors concluded that the continuous 30-day supplementation of *Rhodiola*-caffeine had superior effects on muscle endurance and explosiveness in both animal and human studies compared to using Rhodiola or caffeine individually [59].

### 3.4. BHP of Rhodiola with Cordyceps for Boosting Exercise Performance

The aim of enhancing the physical performance of *Rhodiola* by combining it with *Cordyceps* has not been achieved in all three studies; the efficacy of BHP of *Rhodiola* rosea with *Cordyceps* was inactive compared to placebo [60,61,62,63]. The fixed combination of Rhodiola rosea with Cordyceps has shown no ergogenic effect on oxygen consumption, cycling time, muscle strength, oxygen uptake, and muscle performance during maximal graded tests following 14 days of supplementation. Meanwhile, both *Rhodiola rosea* and *Cordyceps* alone have demonstrated increased muscle fatigue resistance and strength benefits in active men. *Cordyceps* increased aerobic capacity and *Rhodiola* increased performance and time to exhaustion. The combination of *Rhodiola* with *Cordyceps* negatively affects the efficacy of the intervention due to an unknown antagonistic interaction, probably due to the impact of purine alkaloid Cordycepin [77]. Its chemical structure resembles that of the nucleoside adenosine, which is involved in adenosine triphosphate (ATP) metabolism, providing energy in many processes, including muscle contraction.



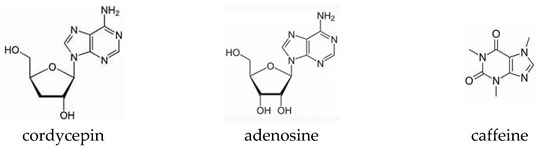



### 3.5. BHP of Rhodiola with Ginkgo biloba for Improvement of Cognitive Function

*Ginkgo biloba* flavones and polyphenols have a potential neuroprotective effect in Alzheimer’s dementia by inhibiting amyloid fibril accumulation [78,79]. *G. biloba* delays age-induced cognitive decline and may have more delicate and subtle therapeutic effects on the speed of cognitive alterations [80]. *G. biloba* is known to improve cognitive impairment and memory, increase cerebral blood circulation in the elderly with dementia syndrome and Alzheimer’s disease, and relieve headaches and migraine [79,80]. Several human clinical trials provide preliminary positive evidence of the antidepressant effects and anxiolytic activity of *G. biloba* [81,82].

The effects of BHP of Arctic root with Ginkgo on the cognitive functions of 112 healthy subjects were recruited in a double-blind, randomized, placebo-controlled trial with four parallel groups. The efficacy of *G. biloba*, *R. rosea* extracts, and their fixed combination was assessed in a psychomotor vigilance task (PVT) and short-term working memory accuracy tests, as compared to placebo effects. The critical flicker-fusion frequency, PVT, and computerized N-back test assessed the central cognitive effect. Short-term working memory accuracy and sustained attention were measured in the psychomotor vigilance task, which provided a numerical measure of sleepiness by counting the number of lapses in the attention of the tested subject. *G. biloba* or *R. rosea* improved PVT and exhibited low to moderate working memory accuracy, whereas BHP exhibited superior effects on PVT, short-term working memory accuracy, and critical fusion versus flicker, compared to G. biloba or *R. rosea* when used alone. The author concluded that combining *R. rosea* with *G. biloba* led to a more significant effect on cognitive performance than either *G. biloba* or *R. rosea* when used alone [64].

### 3.6. BHP of Rhodiola rosea with Actaea racemosa for Relief of Aging-Related Menopausal Symptoms

Aging-related decline in the production of sex hormones is associated with many menopause symptoms, including hot flashes, sweating, sleeplessness, nervousness, irritability, depressive state, palpitation, joint pain, libido changes, vaginal dryness, etc. However, long-term hormonal replacement therapy is known to increase the risk of developing breast cancer. In the meantime, *Rhodiola rosea* is known not only for its stress-protective activity but also for its anti-cancerogenic potential. It was suggested that Rhodiola can counteract menopausal symptoms due to the selective modulation of estrogenic receptors and consequently decrease the risk of cancer in hormone-sensitive tissues during the non-reproductive phase in women, likely due to a decline in the production of sex hormones in menopausal women. Therefore, the BHP (Menopause Relief EP^®^), consisting of a combination of *Rhodiola rosea* (RR) and *Actaea racemosa* (black cohosh) root extracts, compared with the most effective black cohosh (BC) preparation in women with menopausal complaints [65].

A total of 220 women (mean age 52 years) were randomly assigned to receive two capsules of either BC (6.5 mg), BC500 (500 mg), Menopause Relief EP ^®^ (206.5 mg), or placebo once per day for 12 weeks [65]. The efficacy endpoints were relief of menopausal symptoms, including hot flushes, sweating, heart discomfort, sleep problems, joint and muscular discomfort, depressive mood, irritability, anxiety, physical and mental exhaustion, sexual problems, bladder problems, and vaginal dryness. These were measured using the Kupperman Menopausal Index (KMI), the Menopause Relief Score (MRS), and the menopause Utian Quality of Life (UQOL) index. The symptom relief effects of Menopause Relief EP^®^ were significantly superior in all tests to the effects of BC and placebo after their repeated administration for 6 and 12 weeks. Menopause Relief EP^®^ significantly improved the UQOL index in patients, compared to BC, BC500, and placebo, mainly due to the beneficial effects on the emotional and physical health domains (Figure 11). It was concluded that the impact of this BHP is more effective than the mono-drug in relieving menopausal symptoms, particularly psychological symptoms.

### 3.7. BHP of Rhodiola with Saffron in Mild and Moderate Depression

Findings from clinical trials conducted to date indicate that saffron supplementation can improve symptoms of depression in patients with MDD. Saffron was found to be as effective as treatment with conventional antidepressants, such as imipramine and fluoxetine, in randomized, double-blind clinical trials [83]. Supplementation of a BHP of *Rhodiola* with saffron extracts in a daily dose of 308 mg *Rhodiola* and 30 mg *Crocus* for six weeks significantly decreased anxiety and depression scores compared to the baseline in 45 patients with mild-moderate depression. The results of this observational study are encouraging, and serious adverse effects were not recorded. However, a double-blind, randomized study with a positive comparator and placebo control design is needed to confirm these results [66].

### 3.8. BHP of Rhodiola with L-Carnosine in Aging Skin

Carnosine (β-alanine-L-histidine) is an endogenous water-soluble dipeptide. It is mainly distributed in skeletal muscles and its levels depend on age (decreases with age), gender (lower in females), and food (lower in a vegetarian diet). Carnosine plays an essential role in aging-related neurodegeneration, diabetes mellitus, cardiovascular diseases, cancer, and other disorders. The meta-analysis of clinical trial findings indisputably demonstrated a significant beneficial effect of L-carnosine only for diabetes mellitus and cognitive impairment [84]. The pathophysiology of sensitive skin consists of an inflammatory reaction resulting from the abnormal penetration of potentially irritating substances in the skin, which occurs due to skin barrier dysfunction and changes in the production of local neuromodulators. A double-blind comparative study was conducted on 124 volunteers with sensitive skin who received either a formulation containing 1% of *Rhodiola*-L-carnosine BHP or a placebo, applied twice daily for 28 consecutive days [67]. The reduction of transepidermal water loss, positive perceptions of improvements in skin dryness and skin-comfort sensation, and the reduction of discomfort sensation after the stinging test were measured. The *Rhodiola*-L-carnosine BHP treatment produced in vivo protective effects in skin barrier function and a positive subjective response from volunteers with sensitive skin. The authors suggested that the protective effect of *Rhodiola*-L-carnosine BHP on skin barrier function and the positive reaction produced in human subjects with sensitive skin is due to an increase in opioid peptides release, an inhibitory effect on neuropeptides production, and modulation of cytokines production by keratinocytes under ultraviolet stress, as observed in vitro testing [67].

## 4. Critical Appraisal of the Studies and Challenges

### 4.1. The Rationale for Including Caffeine or Caffeine-Containing BHP Based on Rhodiola Extracts

There is an essential difference in pharmacological profiles (signatures) between adaptogens and conventional stimulants, such as caffeine. Caffeine temporarily increases physical and cognitive functions, but unlike adaptogens, it does not exhibit neuro-, hepato-, cardioprotective actions nor does it improve survival under stress. Additionally, caffeine does not induce energy replenishment and is associated with poor quality arousal and recovery process after exhaustive physical load. Notably, unlike *Rhodiola* and other adaptogens, prolonged use of caffeine can cause the user to develop both tolerance and addiction [3].

The magnesium and caffeine content in green tea was uncontrolled and not specified [40,41,42,43]; therefore, there is no guarantee that the effect of Mg-Teadiola will consistently provide reproducible efficacy.

In this context, stern concerns occur regarding the rationale for including caffeine or green tea containing caffeine in the formulation of the safe adaptogenic BHP supplement Mg-Teadiole [40,41,42,43]. Addiction to caffeine and other undesired effects of regular caffeine supplementation are well known. In this context, the formulation’s rationale does not stand up to scrutiny.

Similarly, combining *R. rosea* extracts with caffeine to increase physical performance has no apparent benefits, particularly in the studies [58,59], where the daily dose of caffeine was at the safety threshold—400 mg.

### 4.2. Issues Regarding Product Quality and the Quality of Clinical Trials 

The magnesium and caffeine content in green tea is uncontrolled and not specified; therefore, there is no guarantee that the effect of Mg-Teadiola will provide consistently reproducible efficacy.

The authors declare the labeled amount of active ingredients [40,41,42,43] but do not adhere to Extensions of the Consolidated Standards of Reporting Trials Statement for Herbal Medicinal Interventions (CONSORT) regarding the quality of the product. Mg-Teadiole was not adequately characterized for several factors, including extraction solvents, the dry herb: dry native extract ratio (DER), the content of active markers (caffeine, theanine, Mg^+2^, salidroside, rosavin, etc.), and the analytical methods validated for selectivity, accuracy, and precision. In addition, TLC and HPLC fingerprints were not provided to ensure reproducible quality and reproducible pharmacological activity. The placebo and Mg-Teadiola were distinguishable by appearance. Reporting the masking procedure is not convincing enough to provide an adequate double-blind study design. 

The reporting of *Rhodiola*-caffeine studies did not comply with the requirements outlined in CONSORT, particularly regarding characteristics of the herbal product, e.g.,

The part(s) of the plant used to produce the product or extract;The type and concentration of extraction solvent used;Dry herb: dry native extract ratio (DER);Qualitative testing (product’s chemical fingerprint, HPLC fingerprint);Doses (number of capsules per day);Description and the results of analytical methods validation (selectivity, accuracy, and precision);Sample size (12 subjects per group) justification;Product randomization (who generated the randomization allocation sequence and where the verum and placebo were encoded);Method of randomization (a method used to generate the random allocation sequence, including details of restriction);Allocation concealment;Masking (how the care provider, investigator, and outcomes assessor were blinded to study preparations);Implementation (who enrolled participants and assigned participants to their groups, etc.);Procedure for treatment compliance (how measurements of compliance of individual patients with the treatment regimen under study were documented);Monitoring;Settings and locations where the data were collected;Deviations from the protocol;Quality assurance and quality control;The voucher specimen (i.e., the retention sample was retained and, if so, kept or deposited);Statistical analysis of adverse events (e.g., the odds ratio (OR) statistics);The other measures to exclude the risk of bias were not specified.

### 4.3. Reproducibility and Consistency of the Results of Studies

The reproducible quality of herbal interventions is the primary issue in ensuring the reproducible efficacy of herbal medicines. However, along with the problem of a lack of independent replications, there are other concerns related to poor reporting and methodological quality issues. These include unclear risk of bias, optimized doses, and treatment regimens to rectify and observe the reproducible efficacy in various clinical studies [44,85].

Green tea and Rhodiola rosea products are known as botanicals with significant variability that depends on numerous factors, including genetic, environmental (climate, soil, insects, pest, microbiological infection) factors, processing methods, and storage conditions (extraction solvent polarity, temperature, duration, etc.) [13,14,86]. The content of active ingredients in herbal preparations depends on many factors, including the geographic and climate zone where the herbs were grown, the season and conditions under which they were harvested, and how they were dried, extracted, and prepared to form the final dosage.

## 5. Limitations of the Review

### 5.1. Literature Search Strategy

The studies were selected using the keywords *Rhodiola rosea, network pharmacology, gene expression, Ingenuity pathways IPA, clinical trials, synergistic effect* in PubMed and in our private library. The results were summarized using a simple analytical framework: Search, Appraisal, Synthesis, and Analysis (SALSA).

### 5.2. Appraisal

Some clinical studies, e.g., Teadiola (see Table 2), were designed to compare the efficacy of Arctic root with green tea and black cohosh. These studies show the superior efficacy of BHP compared with mono-drugs and placebo, presumably due to the synergy of BHP’s ingredients [40,41,42,43].

In the clinical study of binary BHP of Rhodiola and black cohosh, BHP was compared only with black cohosh and placebo, but not with Arctic root. BHP’s efficacy was superior to that of black cohosh, suggesting that these ingredients act at least additively. The limitation of this study was the lack of inclusion of one more group—Arctic root.

Other clinical studies had no active control as a mono-drug, except for a placebo, e.g., BHP of Arctic root and cordyceps, which were found inactive in contrast to cordyceps, which was found active in similar clinical trials, suggesting a possible antagonistic mechanism of action between these mono drugs.

### 5.3. Terminology

There are several definitions of synergy [87,88,89,90,91], including two commonly used definitions of synergy in pharmacology and pharmacognosy related to efficacy, effectiveness, and potency of the combinations of two agents with the same pharmacological activity: “A combination of agents that is more effective than is expected from the effectiveness of its constituents is said to show synergy” [88]. In this review, we adhere to the definition of synergy, which is formulated as “two or more agents working together to produce a result not obtainable by any of the agents independently” and can be interpreted as the generation of new pharmacological activity, which is specific only for the combination of two or more agents [87].

## 6. Future Perspectives

There are many challenges in BHP and PHP research, including:(I)The impact of the chemical composition of active compounds (conditional fingerprint) on the pharmacological activity (conditional signature) of the total extract is crucial due to synergistic, potentiating, and antagonistic interactions between the multiple targets of numerous active components of the extracts.(II)The optimal effective and safe therapeutic dose of *Rhodiola rosea* extract should be established, considering the “bell shape” dose–response relationship of *Rhodiola rosea*. Supraphysiological concentrations far exceeding the proposed dose in humans were applied in many in vitro studies. Most studies have ignored this issue; the pharmacological and toxicological data are usually unavailable. The optimal range of doses of adaptogens is like that of the hormetic zone dose–response pattern. The underlying molecular mechanisms of hormesis are not fully understood. The theoretical background of hormesis is related to the hypothesis of interactions of biologically active compounds (ligand/intervention/drug) with two target proteins (receptors) that have functionally opposite responses at different concentrations. It was proposed that a drug acting as a competitive antagonist at either or both of the receptors changes the relationship between the two opposing concentration-effect curves, resulting in potentiation, antagonism, or reversal of the observed effect; the theoretical model suggested that the total effect in the system can be obtained by the algebraic summation of the two effects resulting from the activation of the two opposing receptor populations providing classical hormesis biphasic curve. However, in practice, dose–response patterns are significantly complicated due to many other interactions with:
multiple targets (receptors) of different affinity to the active compound,other regulatory proteins or mediators in the networks involved in the adaptive stress response,feedback down regulations in the molecular signaling pathways and/or,the metabolic transformation of active ligands into metabolites, a secondary ligand, which have different affinity to various receptors of adaptive stress response.
(III)Quality and safety issues related to the content of potentially toxic compounds should be considered. For example, the cyanogenic nitrile Lotaustralin can be highly toxic and thus is encouraged to be subjected to the quality control process. Although Rhodiola is generally safe, two critical points that need to be considered are possible herb-drug interaction, established in some studies, and potentially toxic cyanogenic nitrile lotaustraline detected in some Rhodiola extracts.

## 7. Conclusions

Considering botanicals comprising multi-component active compounds, their interactions result in novel, unexpected pharmacological activity due to their **synergistic and antagonistic** effects. 

The most important finding from BHP’s network pharmacology studies was the evidence supporting the synergistic interaction of its ingredients, revealing unexpected new pharmacological activities unique to and specific to the new BHP.

The results of some studies show the superior efficacy of BHP, such as the combinations of Arctic root with black cohosh, Arctic root with Ashwagandha, Arctic root with green tea, Arctic root with Ginkgo, Arctic root with Eleuthero and Schisandra, compared to the mono-drugs. On the contrary, some a priori designed combinations, such as Rhodiola with Cordyceps, were inactive in several clinical studies, presumably due to antagonistic interactions and crosstalk between molecular targets within the molecular networks involved in the cellular and overall response of organisms to the intervention.

The network pharmacology approach is suitable for understanding their mechanisms of action and predicting possible toxic effects, new indications for use, and lack of activity. However, the results of in silico analysis must be based on experimental findings of gene and protein expression of isolated cells and validated at least through in vivo experiments on rodents.

## Figures and Tables

**Figure 1 pharmaceuticals-17-00483-f001:**
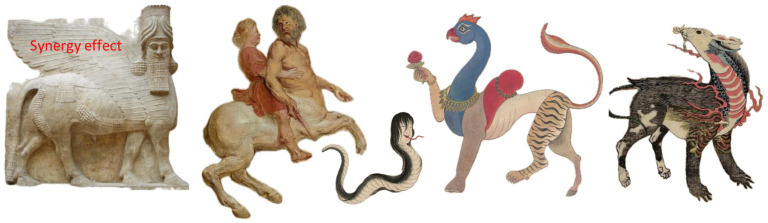
Images of two, three, and more kinds of hybrids from ancient mythology. https://en.wikipedia.org/wiki/List_of_hybrid_creatures_in_folklore, accessed on 3 April 2024.

**Figure 2 pharmaceuticals-17-00483-f002:**
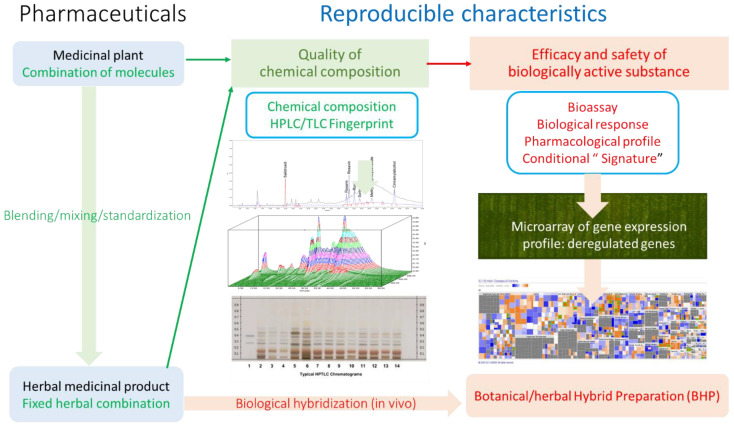
Schematic representation of quality, efficacy, and safety characteristics of herbal medicinal products comprising the mixtures of fixed combinations of molecules from herbal extracts. Reproducible qualitative and quantitative chemical composition by HPLC and TLC fingerprint ensures the reproducible quality of a fixed combination. Reproducible efficacy and safety of a botanical/herbal hybrid preparation (BHP) is characterized by pharmacological profile—conditional signature, e.g., microarray dataset of deregulated genes in response to exposure of BHP in a bioassay providing further information on the effect on physiological functions and diseases in the form of heatmaps.

**Figure 3 pharmaceuticals-17-00483-f003:**
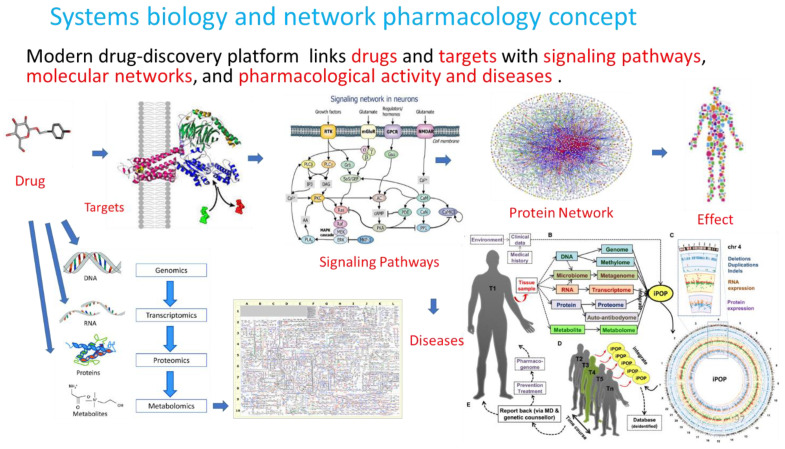
Molecular network-based cause–effect relationships concept. The current drug-discovery platform links drugs and targets with signaling pathways, molecular networks, and organism effects. Molecular targets of adaptogens, their networks, and signaling pathways are associated with chronic inflammation, atherosclerosis, neurodegenerative cognitive impairment, metabolic disorders, and cancer. Predicting the response of the human body to medication requires an understanding of drug–effect relationships at the organism, organ, tissue, cellular, and molecular levels based on integrative personal OMICS (DNA-genomics, RNA-transcriptomics, microbiomes, proteomics, and metabolomics) profiling and their changes in health and disease, as well as after pharmacological intervention.

**Figure 4 pharmaceuticals-17-00483-f004:**
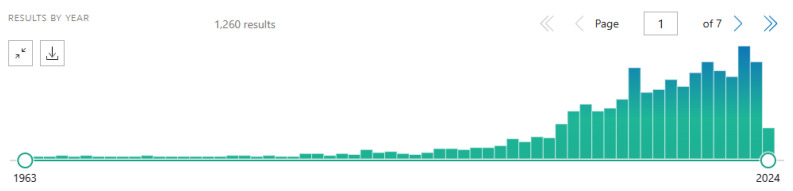
The number of publications (1200 results) worldwide on *R. rosea* from 1963 to 2024; the author’s chart extracted from Pubmed: rhodiola rosea—search results—PubMed (nih.gov, accessed on 3 April 2024).

**Figure 5 pharmaceuticals-17-00483-f005:**
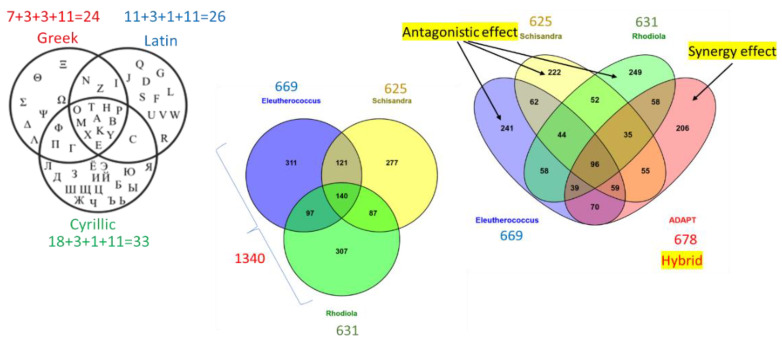
The upper panel shows a Venn diagram with Greek, Latin, and Cyrillic letter symbols, where overlapped sections contain the same symbols. Eleven symbols are the same in all alphabets. Similarly, *Rhodiola*, *Eleutherococcus*, and *Schisandra* separately deregulated 140 out of 1340 genes in experiments with neuroglia cells; however, only 96 out of 640 genes were deregulated by the hybrid combination (ADAPT) of these three plant extracts. External sections of the Venn diagram show the number of deregulated genes specific to distinct extracts (antagonistic interactions in blue, yellow, and green) and synergy-derived 206 deregulated genes characteristic to ADAPT-232 (in red). The lower panel shows the number of compounds in extracts and 3D-HPLC fingerprints of *Rhodiola, Eleutherococcus Schisandra*, and their combination ADAPT-232, and Venn diagrams showing intersections of deregulated genes in neuroglia cells after exposure to these herbal extracts. Authors’ drawings adapted from freely accessible publications [5].

**Figure 6 pharmaceuticals-17-00483-f006:**
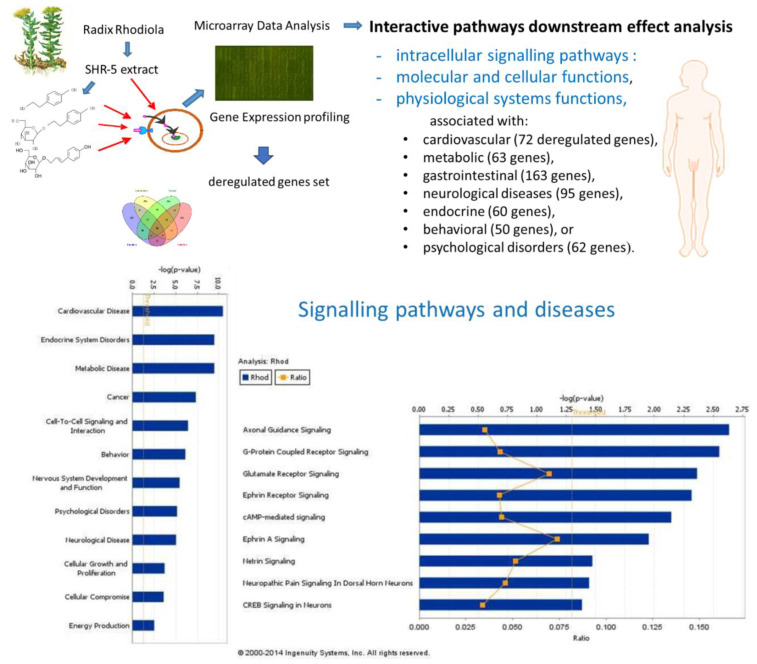
The impact of *Rhodiola* and its active compounds on gene expression in human brain cells suggests potential effects on canonical intracellular signaling pathways, molecular, cellular, and physiological functions, and diseases; updated from the authors’ freely accessible publication [9] and authors’ drawings.

**Figure 7 pharmaceuticals-17-00483-f007:**
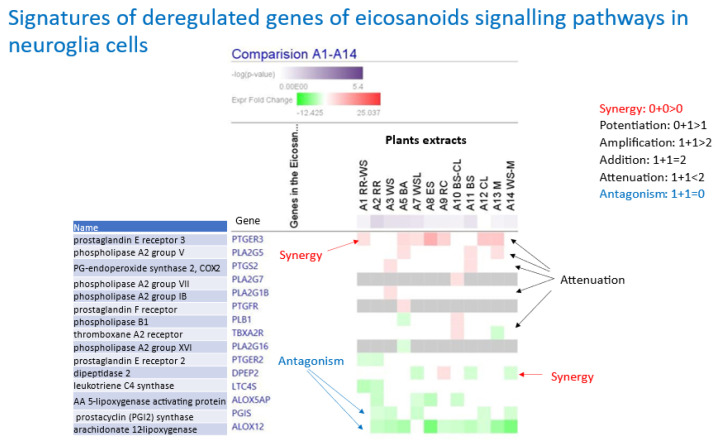
The synergy, potentiation, and antagonistic effects of hybridization of a combination of *Rhodiola* with *Withania*, *Withania* with melatonin, and *Curcuma longa* with *Boswellia* on eicosanoids signaling pathways, which has an essential role in inflammation and neurodegeneration assessed in isolated neuroglia cells. The authors’ drawings were adapted from freely accessible publications [65].

**Figure 8 pharmaceuticals-17-00483-f008:**
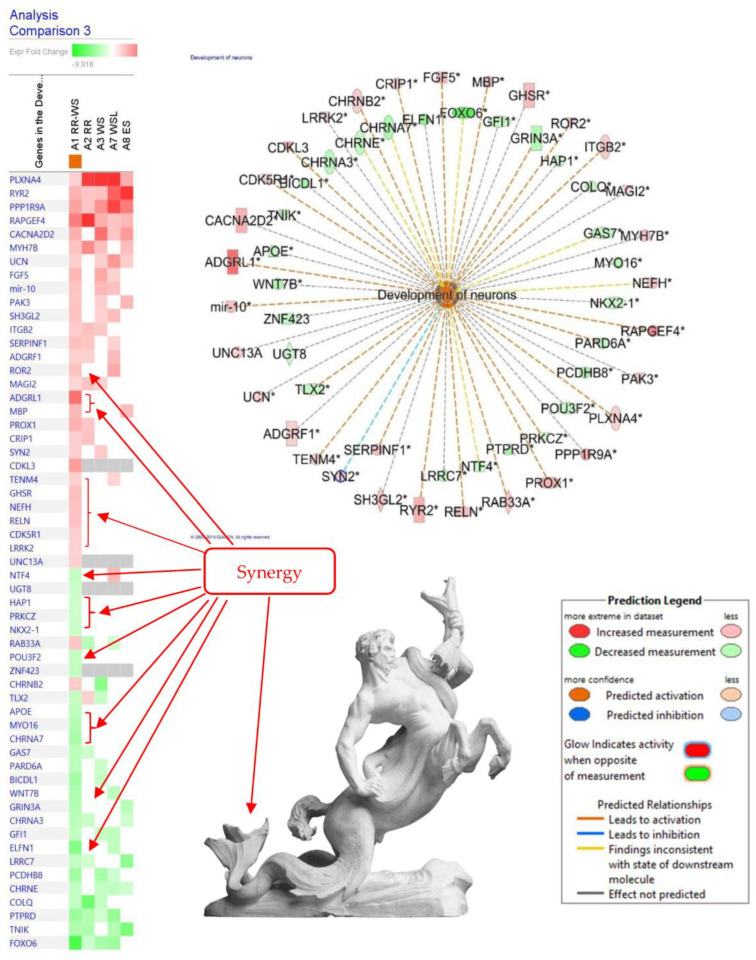
The effects of RR-WS (Adaptra) on gene expression in human T98G neuroglia cells and the predicted activation of the development of neurons. The authors’ drawings were adapted from freely accessible publications [8]. The synergy effects (red arrows) of hybridization of *Rhodiola* with *Withania* on neurogenesis signaling pathways in isolated neuroglia cells. The intensity of green and red squares indicates fold-changes compared to control, where green means down- and red means up-regulation. Synergistic or antagonistic effects on gene expression were observed by comparing the impact of the BHP Adaptra combination of RR-WS (sample A1) with the lack of impact of individual extracts of RR (*R. rosea*), WS (*Withania somnifera*), and WSL *Withania somnifera* low dose, corresponding to samples A2, A3, and A7, at a significance level of *p* < 0.05 (−log = 1.3) and a z-score > 2. The symbolic interpretation of synergy and antagonism by the image of hybrid creature from Greek mythology: *ichthyocentaurs* with a human head, a horse’s body-derived fish-tail due to their synergistic and antagonistic (e.g., lack of human legs) interactions.

**Figure 9 pharmaceuticals-17-00483-f009:**
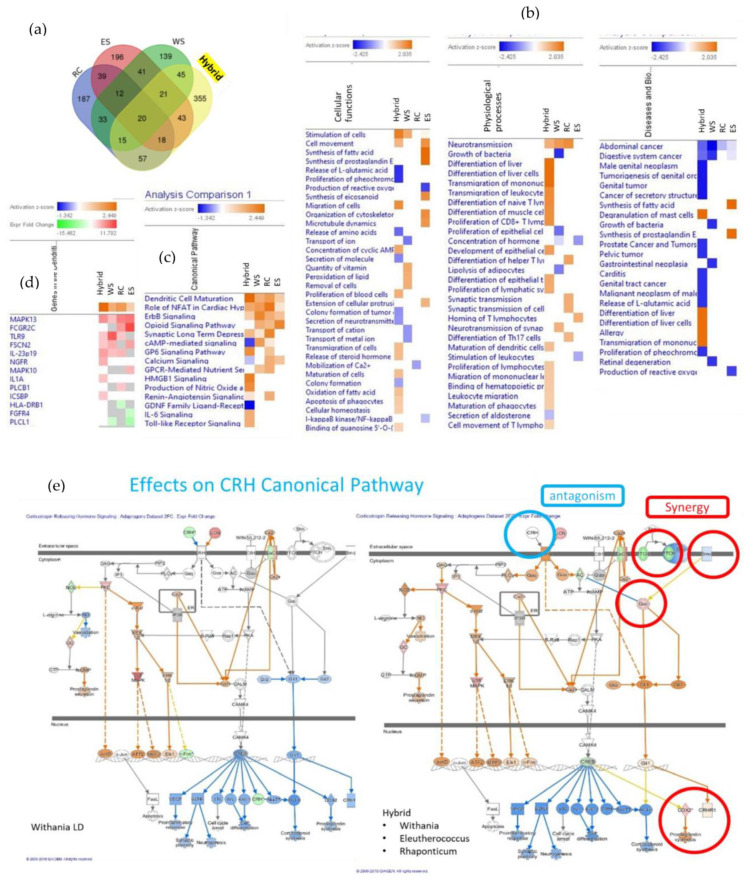
(**a**) Venn diagrams of deregulated genes caused by treatment of neuroglial cells with *Rhaponticum cartamoides* L. (RC), *Eleutherococcus senticosus* (RS), and *Withania somnifera* (WS) root extracts, as well as their hybrid combination (RC-ES-WS). Values show the number of unique genes up- or downregulated by each extract alone and the number of deregulated genes overlapping extracts. (**b**,**c**) Heatmaps of canonical pathways, cellular functions, physiological processes, and diseases activated (brown) and inhibited (blue) by treatment of neuroglial cells with WS, RC, ES, and the hybrid combination RC-ES-WS. (**d**) Heatmap of gene expression showing synergistic or antagonistic effects of the ingredients of BHP RC-ES-WS on dendritic cell maturation pathway. Upregulated genes are shown in red, while downregulated genes are highlighted in green. (**e**) Effects on CFH Canonical Pathway. Authors’ drawings adapted from freely accessible publications [6].

**Figure 10 pharmaceuticals-17-00483-f010:**
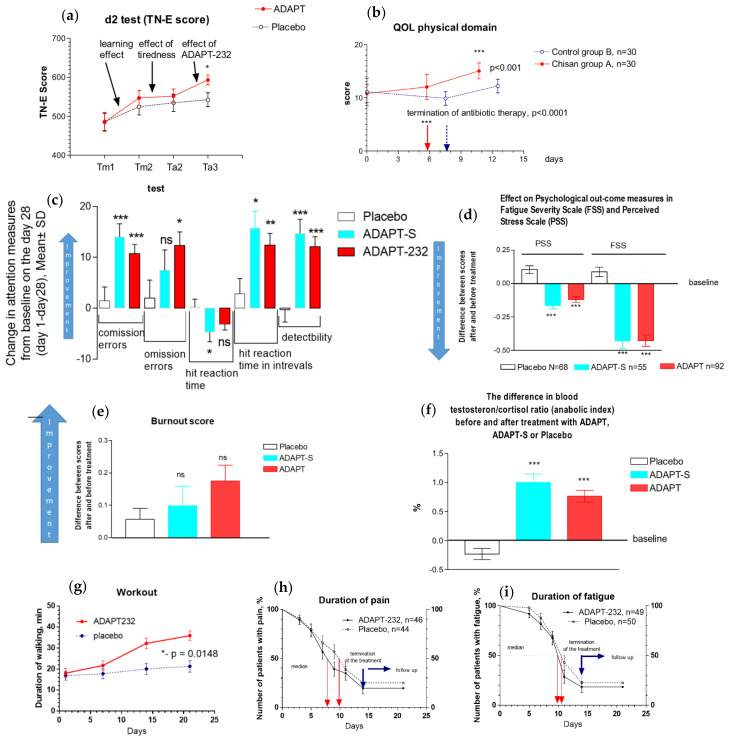
(**a**) Single-dose effects of ADAPT-232 on cognitive functions in participants exposed to stressful cognitive tasks (Stroop CW and d2 tests) four times over three consecutive days (day 1: in the morning, when subjects were not tired; day 2: in the morning, when subjects were not tired and in the afternoon, when subjects were tired; day 3: in the afternoon, two hours after treatment with ADAPT or placebo, when subjects were tired). (**b**)—Effect of ADAPT-232 on physical domain of QOL in patients during their recovery from pneumonia. (**c**–**f**)—Effect of ADAPT-232 and the BHP, containing Rhodiola, Eleutherococcus, Schisandra, and Leuzea (Rhaponticum cartamoides L.) roots extracts, on decreased inattention, impulsivity, and the perception of stress, fatigue and the anabolic index (testosterone/cortisol ratio) in 200 elite athletes. (**g**–**i**)—Effect of ADAPT-232 on the duration of pain, fatigue, and physical activity of patients with long COVID symptoms. *—*p* < 0.05; **—*p* < 0.01; ***—*p* < 0.001, *ns*—not significant difference.

**Figure 11 pharmaceuticals-17-00483-f011:**
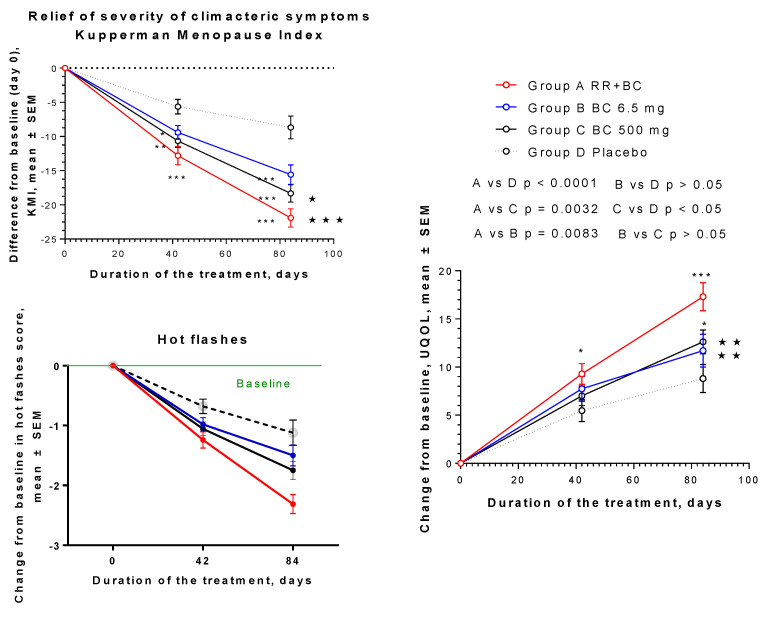
Superior effect of the combination of *Rhodiola* with black cohosh on climacteric symptoms of women and their quality of life. This publication [65] is open access and is distributed under the Creative Commons Attribution License, which permits unrestricted use, distribution, and reproduction in any medium, provided the original work is properly cited. *—*p* < 0.05; **—*p* < 0.01; ***—*p* < 0.001.

**Table 1 pharmaceuticals-17-00483-t001:** Effect of *Rhodiola rosea* (RR), *Withania somnifera* (WS), and their combination RR-WS (BHP Adaptra) on genes involved in the regulation of neuronal development *.

Gene Symbol	Entrez Gene Name	Literature Findings	Prediction **	Gene Expression, Fold Change		
				RR-WS	RR	WS
*ADGRF1*	adhesion G protein-coupled receptor F1	Affects (4)	Affected	2.29	2.28	
* **ADGRL1 ***** *	**adhesion G protein-coupled receptor L1**	Increases (2)	Increased	6.93		
* **APOE** *	**apolipoprotein E**	Affects (13)	Affected	−2.84		
*BICDL1*	BICD family like cargo adaptor 1	Affects (2)	Affected	−3.98		−2.34
*CACNA2D2*	calcium voltage-gated channel auxiliary subunit α2 δ2	Affects (2)	Affected	3.76		6.93
* **CDK5R1** *	**cyclin-dependent kinase 5 regulatory subunit 1**	Increases (4)	Increased	2.33		
* **CDKL3** *	**cyclin-dependent kinase like 3**	Increases (3)	Increased	4.82		
*CHRNA3*	cholinergic receptor nicotinic α 3 subunit	Affects (2)	Affected	−3.09	−2.45	
* **CHRNA7** *	**cholinergic receptor nicotinic α 7 subunit**	Increases (1)	Decreased	−3.74		
*CHRNB2*	cholinergic receptor nicotinic β 2 subunit	Increases (8)	Increased	2.45		−5.20
*CHRNE*	cholinergic receptor nicotinic epsilon subunit	Increases (1)	Decreased	−2.65		−2.59
*COLQ*	collagen-like tail subunit of acetylcholinesterase	Affects (2)	Affected	−2.65	−6.30	−2.69
*CRIP1*	cysteine rich protein 1	Increases (1)	Increased	2.41	3.01	
* **ELFN1** *	**extracellular leucine-rich repeat and fibronectin type III domain-containing 1**	Affects (1)	Affected	−5.31		
*FGF5*	fibroblast growth factor 5	Increases (1)	Increased	3.52		4.23
*FOXO6*	forkhead box O6	Increases (3)	Decreased	−7.93	−2.10	−3.89
*GAS7*	growth arrest specific 7	Increases (3)	Decreased	−2.85	−2.26	
*GFI1*	growth factor independent 1 transcriptional repressor	Affects (1)	Affected	−2.65		−2.59
* **GHSR** *	**growth hormone secretagogue receptor**	Affects (3)	Affected	3.15		
* **GRIN3A** *	**glutamate ionotropic receptor NMDA type subunit 3A**	Decreases (4)	Increased	−3.33		
* **HAP1** *	**huntingtin-associated protein 1**	Affects (1)	Affected	−2.21		
*ITGB2*	integrin subunit β 2	Increases (1)	Increased	2.45	3.05	2.66
*LRRC7*	leucine-rich repeat containing 7	Affects (1)	Affected	−2.66	−2.11	
* **LRRK2** *	**leucine-rich repeat kinase 2**	Affects (4)	Affected	2.26		
*MAGI2*	membrane-associated guanylate kinase,	Affects (10)	Affected	2.01	3.01	2.18
* **MBP** *	**myelin basic protein**	Increases (1)	Increased	3.48		
*mir-10*	microRNA 100	Increases (1)	Increased	2.89		3.53
*MYH7B*	myosin heavy chain 7B	Affects (1)	Affected	3.01	5.70	3.08
* **MYO16** *	**myosin XVI**	Affects (1)	Affected	−3.32		
* **NEFH** *	**neurofilament heavy**	Decreases (18)	Decreased	3.02		
* **NKX2-1** *	**NK2 homeobox 1**	Affects (4)	Affected	−2.38		
* **NTF4** *	**neurotrophin 4**	Increases (5)	Decreased	−2.61		
*PAK3*	p21 (RAC1) activated kinase 3	Affects (4)	Affected	2.86		2.36
*PARD6A*	par-6 family cell polarity regulator α	Decreases (2)	Increased	−2.84		−2.39
*PCDHB8*	protocadherin β 8	Affects (1)	Affected	−3.97		−3.89
*PLXNA4*	plexin A4	Increases (5)	Increased	2.25	9.49	10.97
* **POU3F2** *	**POU class 3 homeobox 2**	Affects (4)	Affected	−2.65		
*PPP1R9A*	protein phosphatase 1 regulatory subunit 9A	Affects (6)	Affected	4.51	2.85	5.38
* **PRKCZ** *	**protein kinase C ζ**	Decreases (2)	Increased	−2.23		
*PROX1*	prospero homeobox 1	Increases (1)	Increased	3.76	2.85	
*PTPRD*	protein tyrosine phosphatase, receptor type D	Increases (3)	Decreased	−4.32	−3.42	−2.11
*RAB33A*	RAB33A, member RAS oncogene family	Increases (1)	Increased	2.81	−3.11	
*RAPGEF4*	Rap guanine nucleotide exchange factor 4	Increases (2)	Increased	6.31	11.27	3.92
* **RELN** *	**reelin**	Increases (9)	Increased	3.01		
* **ROR2** *	**receptor tyrosine kinase-like orphan receptor 2**	Increases (5)	Increased	3.01		
*RYR2*	ryanodine receptor 2	Increases (2)	Increased	3.75	2.85	3.07
*SERPINF1*	serpin family F member 1	Increases (1)	Increased	3.04	2.88	
*SH3GL2*	SH3 domain containing GRB2-like 2, endophilin A1	Affects (2)	Affected	3.02		2.16
*SYN2*	synapsin II	Affects (3)	Affected	2.41		2.56
* **TENM4** *	**teneurin transmembrane protein 4**	Increases (3)	Increased	2.25		
*TLX2*	T cell leukemia homeobox 2	Decreases (2)	Increased	−2.64	2.39	−2.59
*TNIK*	TRAF2 and NCK interacting kinase	Affects (1)	Affected	−3.53	−3.60	
*UCN*	urocortin	Affects (1)	Affected	2.35		2.38
* **UGT8** *	**UDP glycosyltransferase 8**	Affects (2)	Affected	−2.21		
* **UNC13A** *	**unc-13 homolog A**	Affects (2)	Affected	2.25		
* **WNT7B** *	**Wnt family member 7B**	Affects (2)	Affected	−3.54		
* **ZNF423** *	**zinc finger protein 423**	Affects (4)	Affected	−2.66		

* Development of neurons predicted to be increased (z-score −2.87). Overlap *p*-value 7.29 × 10^−3^. ** Prediction is based on measurement direction and literature data: 22 of 57 genes deregulated by RR-WS have measurement direction consistent with an increase in the development of neurons. *** Twenty-five genes (in **bold** text) deregulated due to synergistic interactions of RR and WS in the fixed combination Adaptra are highlighted in red text.

**Table 2 pharmaceuticals-17-00483-t002:** Clinical studies of BHP of Rhodiola with other plants.

Reference/ Year	BHP Name, Ingredients	Condition	Population (n)/ Country	Dosage and Active Markers	Daily Dose and Duration of Treatment	Study Design * and Comparator	Result and Outcomes
Dye et al., 2020 [49]	**Mg-Teadiola**^®^: *Rhodiola rosea* L. + *Camelia chinensis* [L.] Kuntze + Mg + vitamins B6, B9, B12+L-theanine	Acute social stress	100 (25 + 25 + 25 + 25) Healthy, moderately stressed (DASS score: 13–25)	125 mg of IC dry extracts of *Camellia sinensis* L. leaf containing 50 mg L-theanine, and 222 mg of IC *Rhodiola rosea* L. root extract (corresponding to 1887 mg plant), and Mg (150 mg elemental) + vitamins B6 (0.7 mg), B9 (0.1 mg), B12 (0.00125 mg) One tablet of Mg- Teadiola^®^ contains 150 mg of Mg, 0.7 mg of vitamin B6, 0.1 mg of vitamin B9, and 1.25 g of vitamin B12, and 222 mg of *Rhodiola rosea* rhizome dry extract, as well as 125 mg of green tea extract, including 50 mg of L-theanine	Single dose One tablet	DB-R-PC-PG, Placebo Capsules Tablets	Subjective stress (stress and arousal), Mood (profile of mood states) TSST
Boyle et al., 2021 [50]	**Mg-Teadiola** ^®^	Acute social stress	25 + 25 + 25 + 25 Healthy, moderately stressed (DASS score: 13–25)			DB-R-PC-PG, Placebo Capsules tablets	TSST Spectral theta brain activity associated with cognitive task performance Salivary cortisol, cardiovascular parameters (BP, HRV)
Boyle et al., 2022 [51]	**Mg-Teadiola** ^®^	Acute social stress	25 + 25 + 25 + 25 Healthy, moderately stressed (DASS score: 13–25)			DB-R-PC-PG, Placebo Capsules tablets	TSST Spectral theta brain activity, attentional capacity
Noah et al., 2022 [52]	**Mg-Teadiola**^®^:	Chronic negative emotional states	49 + 51 Healthy, moderately stressed (DASS score: >14)		One tablet daily for 28 days	R-PC-PG, Placebo Tablets	Stress, anxiety, depression, sleep, cortisol
Pickering et al., 2022 [53]	**Mg-Teadiola** ^®^	Thermal stimulation	20 + 20 Healthy, moderately stressed (DASS score: >14)			R-PC-PG, Placebo tablets	blood-oxygen-level-dependent (BOLD) signal, stress, anxiety, depression, and sleep, cortisol
Bangratz et al., 2018 [66]	*Rhodiola rosea* + *Crocus sativus* L.	Depression	45	308 mg *Rhodiola* and 30 mg *Crocus*	42 days	OL	
Al-Kuraishy, 2015 [64]	*Rhodiola rosea* + *Ginkgo biloba*	cognitive function	112 (27 + 25 + 30 + 30)	*R. rosea* capsule 500 mg/day, *G. biloba* capsule 60 mg/day (standardized to contain 24% Ginkgo flavone glycosides)	10 days	R-PC-PG, Placebo capsules	Short-term working memory accuracy test (computerized N-back test) psychomotor vigilance task
Liu et al., 2023 [58]	*Rhodiola rosea* + caffeine	physical performance in resistance exercise	48 (12 + 12 + 12 + 12) 24 exercise-trained and 24 untrained healthy subjects	Rhodiola (1.2 g/capsule, 12 mg rhodioloside/salidroside) caffeine (200 mg/capsule; 3 mg/kg)	30 days 2 capsules/day	R-DB-PC-CO (?) Placebo	Significant improvements in muscle strength and muscular endurance compared to the placebo
Yun et al., 2024 [59]	*Rhodiola rosea +* caffeine	muscle endurance and explosiveness in humans	72 (6 groups × 12) 24 exercise-trained and 48 untrained healthy subjects	Rhodiola (1.2 g/capsule) caffeine (200 mg/capsule; 3 mg/kg)	30 days 2 capsules/day	R-OL-PC Placebo	Significant improvements in muscle strength and muscular endurance (5 km run, countermovement jump, standing long jump, 30 m sprint), oxygen consumption (VO2max) compared to the placebo
Earnst et al., 2004 [60]	*Rhodiola rosea* + *Cordyceps sinensis*	Exercise performance	17 healthy subjects	1000 mg *Cordyceps sinensis* + 300 mg RR 3.0% rosavin and 2.5% salidroside	6 capsules/day 4 days, then a maintenance dose of 3 capsules/day for 11 day	R-DB-PC placebo	No significant difference between or within groups
Coulson et al., 2005 [61]	*Rhodiola rosea* + *Cordyceps sinensis*	Exercise performance	8	1000 mg *Cordyceps sinensis* + 300 mg RR 3.0% rosavin and 2.5% salidroside	6 capsules/day 4 days, then a maintenance dose of 3 capsules/day for 7 day	R-DB-PC placebo	After the pre-post endurance test, no significant difference between intervention and placebo in muscle tissue oxygen saturation; no significant (*p* </= 0.05) differences in ventilatory threshold (V(T)) or time to exhaustion (T(E)) between or within the treatment or control group
Kriepke et al., 2020 [63]	*Rhodiola rosea* + *Cordyceps sinensis* + blend of other 11 adaptogens	Exercise performance	10 + 11	NS	14-week	R-DB-PC placebo	No significant difference between or within groups
Dieamant et al., 2008 [67]	*Rhodiola rosea* + L-carnosine	Aging skin	62 + 62	1% of RCAC topical	28 days	DB-PC placebo	Protective effect of RCAC on skin barrier function and the positive response produced in human subjects with sensitive skin
Pkhaladze et al., 2020 [65]	**Menopause Relief EP^®^**: *Rhodiola rosea* EPR-7^®^ (RR) + *Actaea racemose* EP40^®^, (black cohosh, BC)	Menopausal complaints	220 elderly woman (55 + 55 + 55 + 55)	Menopause Relief EP^®^ capsules, 206.5 mg, containing 200 mg *R rosea* rhizome extract EPR-7^®^ and 6.5 mg of *A. racemose* rhizoma dry extract, EP40^®^	2 capsules/day for 84 days	R-DB-PC-PG Placebo BC 6.5 mg BC 300 mg	BC is more effective in combination with RR in the relief of menopausal symptoms, particularly psychological symptoms Kupperman Menopausal Index (KMI), Menopause Relief Score (MRS), and menopause Utian Quality of Life (UQOL) index
Narimanian et al., 2005 [54]	**ADAPT-232 (Chisan^®^):** *Rhodiola rosea +* *Schisandra +* *Eleutherococcus*	Acute nonspecific pneumonia	60 (30 + 30)	BHP of extracts from roots of *R. rosea* L. (27.6%), from berries of *S. chinensis* (51.0%), and from roots of *E. senticosus* (24.4%), standardized to contain 0.068 mg/mL salidroside, 0.141 mg/mL rosavin, 0.177 mg/mL shisandrin, 0.105 mg/mL gamma-shisandrin, and eleutherosides B and E (0.0 11 and 0.027 mg/mL)	40 mL (20 + 20), 10–15 days	R-DB-PC-PG placebo	Adjuvant therapy with ADAPT-232 decreased the duration of patients’ recovery time and the acute phase of the illness; it also increased the mental performance of patients in the rehabilitation period and improved their quality of life (QOL) Duration of antibiotic therapy, psychometric tests, and the QOL
Schutgens et.al., 2009 [18]	**ADAPT-232:** *Rhodiola rosea +* *Schisandra +* *Eleutherococcus*	Ultraweak Biophoton Emission	30 (10 + 10 + 10) Healthy subject experienced levels of stress and of fatigue (tiredness)	One tablet (456 mg) including 140 mg of the proprietary blend ADAPT-232) contains 0.5% schisandrin, 0.47% salidroside, and 0.59% rosavin, 0.11% One Rhodiola tablet (456 mg) including 144 mg SHR-5 extract contains 2.3% salidroside, 0.4% *p*-tyrosol, and 2.7% rosavin	Two tablets 7 days	R-DB-PC-PG Placebo Rhodiola rosea	ADAPT-232 and Rhodiola rosea (SHR-5) were able to reduce photon emission; however, only Rhodiola rosea (SHR-5) significantly reduced photon emission compared with the placebo group; Rhodiola, but not ADAPT-232, reduced fatigue
Aslanyan et al., 2010 [55]	**ADAPT-232 (Chisan^®^):** *Rhodiola rosea +* *Schisandra +* *Eleutherococcus*	Stressful cognitive tasks (Stroop Color-Word test and the d2Test of attention, fatigue	40 (20 + 20) Healthy women felt stressed over a long period of time by virtue of living under psychologically stressful conditions	One capsule of ADAPT-232S contained 0.5 mg of salidroside, 1.0 mg pf Schizandrin, and 0.35 mg of Eleutherosides B and E	Single dose One tablet	R-DB-PC-PG placebo	Significant improvement in attention and increase in speed and accuracy during stressful cognitive tasks in comparison to placebo Mental performance (attention, speed, and accuracy), arterial blood pressure, and heart rate
Karosanidze et al., 2022 [57]	**ADAPT-232 (Chisan^®^):** *Rhodiola rosea +* *Schisandra +* *Eleutherococcus*	Long COVID-19	100 (50 + 50) patients with Long COVID symptoms	One daily dose (2 × 30 mL oral solution) contains 180 mg extract of *R. rosea* rhizome, 600 mg of *S.chinensis* berry, and 156 mg of *E. senticosus* radix extracts	60 mL 30 days	R-QB-PC-PG Placebo	There was a significant increase in physical performance and recovery in long-term COVID patients; the duration of fatigue and chronic pain decreased; and the severity of all long-term COVID symptoms was relieved Duration of symptoms of long COVID
Hovhannisyan et al., 2015 [56]	**ADAPT-232S:** *Rhodiola rosea +* *Schisandra +* *Eleutherococcus*	exercise performance	215 (92 + 55 + 68) healthy athletes aged 18–35	One capsule of ADAPT-232S contains 0.5 mg of salidroside, 1.0 mg pf Schizandrin, and 0.35 mg of Eleutherosides B and E	2 capsules × 2 times a day, 30 days	R-DB-PC-PG Placebo	ADAPT-S and ADAPT-232S increase physical performance and the recovery of athletes after heavy physical and emotional loads They significantly decrease inattention, impulsivity, and the perception of stress, reduce fatigue, increase the anabolic index, and have excellent tolerability profiles
	**ADAPT-S:** *Rhodiola rosea +* *Schisandra +* *Eleutherococcus +* *Rhaponticum*			One capsule of ADAPT-S contains 1.5 mg of salidroside, 1.0 mg of Schizandrin, and 0.35 mg of Eleutherosides B and E, as well as 1.5 mg of 20-hydroxyecdisterone			The effects of ADAPT-S were superior with respect to the anabolic index, blood testosterone, and physical performance index; the results of this study suggest that ADAPT-232S and ADAPT-S might be useful for athletes’ recovery after exercising and for preventing the symptoms of overtraining; ADAPT-S was most effective in sports disciplines where high coordination during physical fatigue (wrestling and long jump) is essentially required

* R—randomized, OL—open-label, PC—placebo-controlled, DB—double-blind, SA—single-arm, PG—parallel groups, CO—crossover, SC—sufficiently characterized herbal preparations, IC—insufficiently or purely characterized herbal preparations, NS—not specified.

## Data Availability

Data sharing is not applicable.

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
