# Peer review of "State-of-the-Art Review on Botanical Hybrid Preparations in Phytomedicine and Phytotherapy Research: Background and Perspectives"

_pharmaceuticals, 2024, doi:10.3390/ph17040483_

Round 1
Reviewer 1 Report
Comments and Suggestions for Authors
Suggestions indicated in the pdf file of the text.

Reviewer 2 Report
Comments and Suggestions for Authors
Dear authors,
Having read the submitted article, I have a few observations that could improve its quality.
1.The title of the article should not contain abbreviations.
2.Latin names of plants should be given in full, indicating the species. For example, Ginkgo biloba L., etc.
3. Figure 4 is blurred. Its quality should be improved.
4. Could you please elaborate on how these clinical and network pharmacology studies examining the effects of a herbal preparation (BHP) with Rhodiola and other plants were selected and summarised?
5. Could you please provide more details on the methodology and results of each of these studies so that readers can better understand the content and key points of the findings?
6. Could you explain how the effects of BHP on each of the plants listed, including Rhodiola ..., were assessed and summarised in the clinical trials?
7.Could you provide more information on how these particular herbs were chosen and how they were combined with Rhodiola... to investigate their synergistic effects or possible interactions?
-
8. Could you provide more detailed information about the composition and dosages of the BHS Mg-Teadiola® tablets used in the study to clarify how the effects of this complex were evaluated?
-
9. Could you share more detailed research results related to subjective perception of stress, mood, cortisol levels, and other measurements to better understand how BHS Mg-Teadiola® performed?
-
10. Could you discuss potential limitations or constraints in these studies, as well as future directions for improving the use of BHS Mg-Teadiola® or further research to better understand its efficacy and safety?
- 11. I suggest to revise section 3.3 to include more detailed information on the methodology and results, such as more specific dosages, measurement methods used and results obtained, to clarify how the effects of the combination of Rhodiola and caffeine on muscle strength and endurance were assessed. This will help readers to better understand the results of the study and their clinical relevance.
Comments on the Quality of English Language
Moderate editing of English language required.
Reviewer 3 Report
Comments and Suggestions for Authors
Overall it ids a very nice compilation and presentation. I strongly believe that this review work will draw public attention and will help further in the field of phytotherapy research. Here are some recommendations to upgrade the overall quality of the manuscript:
1. Please include a "Artcle search strategy" section and provide PRISMA model for a clear vision about inclusion and exclusion criteria
2. What does the red ink mean in Table 1?
3. Please provide "Limitations of the study", "Future prospects" in two separate section.
4. Please provide some figures explaining the pharmacological potentials with molecular vision.
After these corrections, the article should be accepted which can coribute in scientific community
Comments on the Quality of English LanguageMinor editing of English language required
Round 2
Reviewer 2 Report
Comments and Suggestions for Authors
The authors have revised the article in line with the comments made, and I recommend accepting it for publication.